# Infrared Imaging of the Brain-Eyelid Thermal Tunnel: A Promising Method for Measuring Body Temperature in Afebrile Children

**DOI:** 10.3390/ijerph20196867

**Published:** 2023-09-30

**Authors:** Franciele De Meneck, Vinicius Santana, Gabriel Carneiro Brioschi, Denise Sabbagh Haddad, Eduardo Borba Neves, Maria do Carmo Franco, Marcos Leal Brioschi

**Affiliations:** 1Division of Nephrology, School of Medicine, Federal University of São Paulo (UNIFESP), São Paulo 04023-062, SP, Brazil; 2Division of Psychiatry, School of Medicine, Federal University of São Paulo (UNIFESP), São Paulo 04023-062, SP, Brazil; vsantana@gmail.com; 3Greenways Academy School, St. Louis, MO 63141, USA; gabrielbrioschi1@gmail.com; 4Faculty of Dentistry, University of São Paulo (FOUSP), São Paulo 05508-000, SP, Brazil; deniseshaddad@hotmail.com; 5Graduate Program in Biomedical Engineering, Federal Technological University of Paraná (UTFPR), Curitiba 80230-901, PR, Brazil; eduardoneves@utfpr.edu.br; 6School of Medicine, Federal University of São Paulo (UNIFESP), São Paulo 04023-062, SP, Brazil; maria.franco@unifesp.br; 7Faculty of Medicine, University of São Paulo (HCFMUSP), São Paulo 05508-000, SP, Brazil

**Keywords:** body temperature, brain, central nervous system, diagnosis, diagnostic techniques and procedures, investigative techniques, pediatrics, thermography, thermometry

## Abstract

(1) Infrared thermography of the inner canthus of the eye has emerged as a promising tool for temperature screening and fever diagnosis. Its non-invasive nature lends itself well to mass screening in diverse settings such as schools, public transport, and healthcare facilities. Swift and accurate temperature assessment plays a pivotal role in the early identification of potential fever cases, facilitating timely isolation, testing, and treatment, thereby mitigating the risk of disease transmission. Nonetheless, the reliability of this approach in the pediatric population, especially when compared to conventional thermometry methods, remains unexplored. This preliminary study aimed to evaluate the concordance between the temperature of the inner canthus of the eye (T_ic,eye_), referred to as the brain-eyelid thermal tunnel (BTT°), with axillary and tympanic methods in afebrile children. (2) Methods: A cohort of 36 children, matched in a 1:1 ratio for gender and age, underwent comprehensive assessments encompassing anthropometric data, blood pressure evaluations, axillary (T_ax_) and tympanic (T_ty_) temperature measurements, as well as BTT° infrared thermography. (3) Results: The findings revealed a high level of concordance among the tympanic, axillary, and BTT° measurement methods. Bland–Altman plots showed that the bias was minimal, and no statistically significant differences were observed when comparing BTT° with axillary (*p* = 0.136) and tympanic (*p* = 0.268) measurements. Passing–Bablok regression scatter plots further confirmed the agreement, aligning the fitted regression line closely with the identity line for both axillary versus BTT° and tympanic (T_ty_) versus BTT° comparisons. (4) Conclusions: This study holds significant implications for public health, especially in the context of infectious disease outbreaks such as COVID-19. BTT° infrared thermography of the inner canthus of the eye (T_ic,eye_) reliably measures body temperature in afebrile children in controlled settings; nevertheless, its practical application necessitates the adaptation of biothermodynamic parameters to accommodate diverse environmental conditions.

## 1. Introduction

Body temperature (T_b_) is widely recognized as one of the most essential vital signs, and the definition of normal T_b_ is used as a reference to help determine the presence of pathological conditions [1]. Currently, the reference established for normothermia is 36.8 °C. However, previous research has indicated that T_b_ is a non-linear parameter that varies according to age, gender, biological rhythm, and ambient temperature [2,3]. While rectal measurement is considered the “gold standard” for measuring T_b_ in children, its invasive nature and potential for causing anxiety limit its application in this population, making alternative techniques more suitable for research purposes [2].

Numerous studies have been conducted to determine the normal range of T_b_ in children by measuring tympanic (T_ty_) and axillary (T_ax_) temperatures. A study involving 301 children with ages ranging from 4.5 months to 14 years (average age = 3.9 ± 3.31 years) reported a mean axillary temperature (T_ax,avg_) of 36.9 °C (range: 34.9–39.8 °C) and an average tympanic temperature (T_ty,avg_) of 36.89 °C (range: 35.3–40.4 °C) [4]. In another study, researchers measured the T_b_ of 1364 infants with a mean age of 72.5 months using the tympanic and axillary methods. They reported that the mean axillary temperature (T_ax_) was 36.0 °C and the average tympanic temperature (T_ty,avg_) was 36.9 °C, with significant differences observed between genders [5].

T_b_ is crucial for maintaining homeostasis, and it is controlled by specific pathways in the brain that operate through thermoregulatory centers located in the hypothalamus [6]. This brain region is recognized as the central controller of thermoregulation in the human body, being responsible for regulating thermoregulatory impulses that arise throughout the body [6].

In the medical field, knowing when the temperature of the brain rises is critical. Although body temperature is thought to be a good predictor of brain temperature, it is not always accurate. As a result, the direct monitoring of brain temperature is advised in critical cases, particularly in pediatric brain injuries. When the patient leaves the intensive care unit, it is common to take the patient’s temperature again in locations such as the rectum, ear, mouth, or eardrum. Infrared thermometry, on the other hand, has been shown in studies to be a useful alternative for measuring brain temperature. Using an infrared thermometer to measure temporal artery temperature, the results are very close to brain temperature, with an average difference of only 0.3 °C. This means that measuring temporal artery temperature can provide a good indication of brain temperature. The inner corner of the eye is another interesting place to estimate core temperature. There are many blood vessels close to the skin’s surface in this region, which can help with estimating body temperature. The medial canthus plays an important role in eyelid function by acting as a fixed-point fulcrum. Because of its close proximity to the arterial supply at the skin’s surface, the inner canthus has been used in mass fever scanning during pandemics such as SARS. Branches of the ophthalmic, dorsal nasal, supratrochlear, superior palpebral, and angular arteries can be found within this small region of the inner canthus. The angular artery, which has three branches, originates from the ophthalmic artery (specifically the supratrochlear branch), and is located directly beneath the skin of the inner canthus. Because of the abundant vascular supply and consequent blood perfusion in this area, it has the potential to be a useful location for estimating core temperature [7,8,9].

Over the past twenty years, the brain thermal tunnel (BTT) has been identified and described as a distinct pathway between the cavernous sinus and the superior medial orbit and the ophthalmic artery and vein, providing a direct and continuous non-invasive method for measuring internal brain temperature [10]. It is acknowledged that the BTT has a thermodynamic structure that transmits a thermal signal, which serves as an indicator of undisturbed thermal transmission from the brain [11]. The surface of the skin overlying the BTT and the superior palpebral vein region is thin dermis (~900 μm) and is fat-free, which results in a lack of thermal insulation. This allows the surface temperature to have a higher intensity infrared emission in this area due to heat transfer from the underlying brain tunnel, thus enabling the assessment of brain temperature [10,12]. An evaluation of this kind is typically conducted by measuring the temperature of the inner canthus of the eye (T_ic,eye_), as the brain-eyelid thermal tunnel (BTT°) overlies the superomedial orbit imaging window [12,13].

Using an infrared thermography method, Haddadin et al. [10,14] demonstrated that BTT° measurements are more accurate than the temperature of the frontal region of the face and other parts of the body. Although validation tests with an infrared thermography method with a measurement uncertainty below ±2% was not conducted in this study, it is worth noting that they reported the reliability, reproducibility, and comparability of infrared thermography images of the BTT to axillary and tympanic procedures for body temperature measurements in young adults. Furthermore, it has been observed that the BTT° is not an effective predictor of body temperature in adults during exercise and during the hyperthermia recovery phase [15]. Currently, the literature is scarce regarding the correlation of the BTT° method with other commonly methods for measuring T_b_ in children, specifically in the context of febrile children [16].

This study aimed to assess the correlation between brain thermal tunnel (BTT°) temperatures, using an infrared thermography method, and axillary (T_ax_) and tympanic (T_ty_) temperatures in afebrile children. This research is of significance as it addresses the need for a non-invasive and reliable method of measuring body temperature in children. The utilization of an infrared thermography method to measure BTT° could potentially enhance temperature screening and fever diagnosis in pediatric populations, as well as serve as a valuable clinical tool for estimating brain temperature in cases of severe intracerebral brain trauma.

## 2. Materials and Methods

This cross-sectional study was conducted with a sample of 36 healthy children aged 8–9 years (18 males and 18 females) enrolled in a low-income public school located near the Federal University of São Paulo (UNIFESP, São Paulo, SP, Brazil) between April and May 2019. None of the children had been diagnosed with a cardiovascular disease, diabetes mellitus, or were actively taking medications. The study was designed and performed in accordance with the Declaration of Helsinki and was approved by the Research Ethics Committee of the Federal University of São Paulo.

Written informed consent was obtained from the parents or guardians of all participants. The sample size was determined using GPower software (version 3.1.9.2; Franz Faul, Unioversität Kiel, Germany).

### 2.1. Clinical Evaluation

Body weight was measured in light clothing without shoes using a portable digital platform scale, while height was measured using a portable stadiometer. Body Mass Index (BMI) was then computed using the formula weight (kg)/height (m^2^). Blood pressure levels were measured on the right arm in a seated position using an automated oscillometric device (Omron HBP1100; Omron Healthcare, Sunrise, FL 33323, USA) with an appropriate cuff size. The blood pressure value was recorded as the average of three measurements taken at two-minute intervals.

### 2.2. Axillary (T_ax,b_) and Infrared Tympanic (T_ty,b_) Temperature Measurement

All children rested for 30 minutes and wore indoor clothes before taking their temperature. The axillary temperature (T_ax_) was measured using a digital thermometer (range: 32–44.9 °C; accuracy: ±0.1 °C; Geratherm Clinic, Germany), while the tympanic temperature (T_ty_) was measured using an infrared thermometer (range: 34–42.2 °C; accuracy: ±0.1 °C; MC-505 Gentle Temp Model, Omron Healthcare, USA). Three measurements were taken on the right side for the axillary (T_ax,b_) and tympanic temperature (T_ty,b_) measurements, with no time gap between the subsequent measurements. These measurements were conducted by the same nurse, who had been trained before the study. The average of the three values was used as the final temperature, and all of the temperature measurements (T_b_) were performed between 13:30 h and 15:30 h. The thermometers were calibrated on a weekly basis to ensure accuracy. The axillary thermometer was immersed in a temperature-controlled water bath set to 37 °C (98.6 °F), which mirrored the standard temperature of the human body. The readings from the thermometer were then compared to the precise water bath temperature and any discrepancies were promptly corrected to ensure accuracy. Similarly, for the tympanic thermometer, calibration was performed within a controlled 35 °C (95 °F) environment, which approximated the typical temperature of the ear canal. Adjustments were made as necessary to maintain precision.

### 2.3. Evaluation of BTT° by Infrared Thermography

The BTT° evaluation was conducted in an acclimatized room with an ambient temperature of T_∞_ ± 2*σ*_T∞_ = 23 ± 1 °C, and a controlled air relative humidity of φ ± 2*σ*_φ_ = 0.6 (60%). The children were seated on a bench in an upright posture, remaining still with their faces parallel to the surface of the bench to maintain a stable position while facing the opposing wall. The children were asked to remove all facial obstructions (e.g., glasses, front hair fringe, garments).

The children rested for 15 minutes within the study area uncovered to achieve thermalization. To avoid heat loss through forced convection, there was no airflow (v_air_ < 0.2 m/s^−1^—surrounding air speed). In order to ensure the accuracy of the results, the children were instructed to avoid consuming caffeine or medications 24 h prior to the evaluation, and to not take hot baths or perform physical exercise less than 2 h before the evaluation.

Thermal images were acquired using an uncooled vanadium oxide microbolometer detector (T430sc, FLIR, Danderyd, Sweden). The detector had a Focal Plane Array (FPA) sensor size of 320 × 240 and operated in the long wavelength spectral band of 7.5–13 μm. It had a focal length of 18 mm, a frame rate of 60 Hz, and a Non-Equivalent Temperature Difference (NEDT) of less than 30 mK (0.03 °C) at 30 °C. Non-Equivalent Temperature Difference (NETD) is a crucial performance metric in thermal imaging equipment, representing its ability to detect small temperature variations. In contrast to thermal sensitivity, NETD takes into account the non-uniformity and variations in the sensor’s response across the image, providing a more accurate measure of the camera’s performance. The detector pitch was 25 μm, and the field of view (FOV) was 25° × 18.8° (IFOV 1.38 mrad). The sensor was calibrated for the evaluation of the temperature range of the human body, with a measurement range of 0 °C to +40 °C (TermoCam^®^ (Honolulu, HI, USA), medical version). The thermal imager was tested using high-quality calibration sources called black bodies. These black bodies had calibrated radiance temperatures with an uncertainty of no more than 0.1 °C (with a confidence level of approximately 95%) and a stability better than ±0.002 °C. The calibration process was carried out in a competent laboratory specializing in radiation thermometry and following international measurement standards. The service centers involved in the calibration were certified according to ISO 9001, and the temperature reference standards were traceable to either the SP Technical Research Institute of Sweden or the National Institute of Standards and Technology (NIST) in the United States. To calibrate the thermal imager, radiation sources were used that could be traced back to the National Standards at RISE, Research Institutes of Sweden, and the NIST. The black body sources used had a range of radiance temperatures and control intervals suitable for laboratory testing as per the specified standard. These black bodies had a known emissivity greater than 0.995. The aperture size of the black body source was sufficiently large to ensure that it did not affect the thermal imager’s temperature measurement and allow for the clear identification of color changes at the workable plane.

The camera’s emissivity was set to ε = 0.95 to accurately measure the temperature of the inner canthus of the radiant eyelid (T_ic,eye_) on both sides (Figure 1), considering that the thinner skin, with a high-k thermal conductivity (k = 0.00009 Kcal/(s·N·C)) and lower fat content in this area [9], typically results in an emissivity value closer to 0.95. All images were taken with the camera at a minimum focus distance of 0.3 meters (0.984 feet) in the multispectral mode. The camera is equipped with an integrated visible-light (optical) camera, which enables users to capture both thermal and visual images. This feature allows for a more accurate determination of the location and precise measurement of the BTT°, facilitating easier analysis and improved documentation.

To ensure accurate temperature measurements of the inner canthus of the radiant eyelid on the children’s faces, the study utilized a carefully calibrated tripod system with built-in leveling mechanisms. The thermographic camera operator was positioned perpendicular to the child’s face plan (90°) and meticulously adjusted the tripod to align the camera’s lens accordingly [17]. The operator received training to maintain this alignment throughout the data acquisition process by monitoring the camera’s position and making real-time adjustments as necessary. This meticulous approach was crucial to minimize measurement errors caused by oblique angles between the camera and the target area, thus ensuring the reliability of temperature measurements. In addition, a specialized trigger mechanism synchronized the thermographic camera’s image acquisition with the child’s natural blinking pattern. The synchronization ensured that the measurements were taken during moments when the child’s eyes were in a stable and relaxed state, minimizing the potential for artifacts or inaccuracies in the data. The trigger mechanism was carefully calibrated to avoid any interference with the child’s comfort or natural behavior during the measurements.

Following the acquisition of the infrared thermography images, the region of interest within the eye was meticulously highlighted manually using an ellipse tool available in the software employed (FLIR ResearchIR Max software). The determination of the maximal BTT° image on both sides (T_ic,eye,max_) was performed by the same certified researcher, who is skilled in infrared thermography imaging and one of the authors of this paper (F.D.M).

### 2.4. Statistical Analysis

Categorical variables were described as frequencies and analyzed using the chi-square test. Numerical variables were examined for normality using the Shapiro–Wilk test and summarized as median, interquartile range, and minimum–maximum values. The Lin´s concordance correlation coefficient (CCC) was used as a measure of precision and accuracy, serving as an indicator of the strength of agreement between methods. The CCC (ρ_c_) assesses the conformity of bivariate observations relative to a gold standard or another reference set. Lin’s CCC (ρc) measures both precision (ρ) and accuracy (Cβ), encompassing a range from 0 to ±1, which is similar to Pearson’s correlation coefficient. According to Altman´s suggestion, it should be interpreted closely with other correlation coefficients, with values <0.2 as poor and >0.8 as excellent [18]. In this study, the agreement was categorized as poor (CCC < 0.90), moderate (CCC = 0.90–0.94), substantial (CCC = 0.95–0.99), or near-perfect (CCC > 0.99). The Passing–Bablok regression analysis and the Bland–Altman method were used to assess the concordance between the three methods. All analyses were conducted using MedCalc Statistical Software version 16.1 (Ostend, Belgium).

## 3. Results

The study enrolled a total of 36 afebrile children, who were paired in a 1:1 ratio based on gender and age (Table 1). The BMI distribution and blood pressure levels were comparable between the two groups. Additionally, the axillary (T_ax_) and tympanic (T_ty_) body temperatures were similar in both genders (Table 1). No significant differences were found between both genders when considering the BTT° temperature (Table 1). Consequently, the following analyses were conducted using the entire population.

Table 2 summarizes the results of the CCC analysis for the comparison of the three methods for the entire cohort. Moderate agreement was observed for all the comparisons (Table 2). Table 3 and Table 4 present the results of the Passing–Bablok regression and Bland–Altman analysis from the comparison between the T_b_ measurement methods. Moreover, Figure 2 shows the plots for the Passing–Bablok regression (dashed and solid lines indicating the line of identity and the best fit, respectively) and the Bland–Altman analysis (dashed lines indicating the upper and lower limits of agreement, and the solid line indicating the bias).

### 3.1. Axillary vs. Tympanic T_b_

The Passing–Bablok regression scatter plots indicated an agreement between the fitted regression line and the identity line (Figure 2A). Furthermore, there was no evidence of constant or proportional differences between the T_b_ measured by the axillary and tympanic methods (Table 3). The Bland–Altman analysis revealed no significant systematic bias between the axillary and tympanic T_b_ methods (*p* = 0.530), with the 95% limits of agreement within the limits of acceptance for both the right and left sides (Table 4); thus, no proportional bias was observed (*p* = 0.364) (Figure 2A).

**Table 3 ijerph-20-06867-t003:** Intercept, slope, and residual standard deviation of Passing–Bablok regression from comparisons between the axillary, tympanic, and BTT° methods.

		Intercept (95% CI) Constant Bias	Slope (95% CI) Proportional Bias	RSD (95% CI)	CUSUM Linearity
Tympanic (°C)	Axillary(°C) BTT° (°C)	0.01 (−0.06 to 0.03) 0.00 (0.00 to 8.38)	1.0 (0.83 to 1.02) 1.0 (0.77 to 1.04)	0.09 (−0.18 to 0.18) 0.10 (−0.19 to 0.19)	0.26 0.54
Tympanic (°C)	BTT° (°C)	0.00 (−3.32 to 2.6 × 10^−12^)	1.0 (1.00 to 1.09)	0.08 (−0.15 to 0.15)	0.99

BTT°: Temperature of the inner canthus of the eye, T_ic,eye_ (brain thermal tunnel); 95% CI: 95% confidence intervals; RSD: Residual Standard Deviation.

**Table 4 ijerph-20-06867-t004:** Bias, Lower and Upper Limits of Agreement from the Bland–Altman plots from the comparisons between the axillary (T_ax_), tympanic (T_ty_), and BTT° methods.

		Tympanic (°C)		Tympanic (°C)
		Axillary (°C)	BTT° (°C)	BTT° (°C)
Bias	(95% CI)	0.01 (−0.06 to 0.03)	0.03 (−0.01 to 0.08)	0.02 (−0.02 to 0.06)
Lower Limit of Agreement	(95% CI)	−0.26 (−0.32 to −0.19)	−0.22 (−0.29 to −0.14)	−0.18 (−0.24 to −0.12)
Upper Limit of Agreement	(95% CI)	0.24 (0.16 to 0.31)	0.28 (0.21 to 0.36)	0.22 (0.16 to 0.28)

BTT°: Temperature of the inner canthus of the eye, T_ic,eye_ (brain thermal tunnel); 95% CI: 95% confidence intervals.

### 3.2. Axillary (T_ax,b_) and Tympanic (T_ax,b_) vs. BTT°

The Passing–Bablok regression scatter plots indicated an agreement of fitted regression line and identity line between the axillary (T_ax_) and BTT° (Figure 2B) methods and between the tympanic and BTT° (Figure 2C) methods. No proportional difference was observed between the axillary (T_ax_) and BTT° or tympanic (T_ty_) and BTT° methods, as evidenced by the results of Table 3. Good concordance between these methods has also been described. The Bland–Altman plots revealed that the bias was close to zero and not statistically significant for the comparison of the BTT° method with the axillary (T_ax_) (*p* = 0.136) (Figure 2B) and tympanic (T_ty_) (*p* = 0.268) methods (Figure 2C) (Table 4). Furthermore, no proportional bias was evident in the Bland–Altman plots (Table 4) (Figure 2B,C).

## 4. Discussion

The current study verified whether the BTT° brain thermal tunnel could provide a valid measure of body temperature, T_b_, in afebrile children. Results showed that there were no substantial differences between axillary (T_ax_) and tympanic (T_ty_) temperatures and BTT° temperatures measured using the infrared thermography method. The Bland–Altman and Passing–Bablok regression plots indicated that the BTT° method was concordant with the traditional axillary and tympanic methods of T_b_ measurement without proportional bias. 

The findings of this study have substantial implications for public health, especially in the context of infectious disease outbreaks like COVID-19. The reliable and accurate measurement of body temperature is essential for identifying fever, a key symptom of various infections. By demonstrating the concordance of BTT° with axillary and tympanic methods, this study establishes the BTT° method as a valuable alternative for measuring body temperature in children. The non-invasive nature of the infrared thermography method makes it suitable for mass screening and monitoring in various settings, such as schools, public transport, and healthcare facilities. Rapid and accurate temperature assessment can help identify potential cases of fever early on, enabling timely isolation, testing, and treatment, thus reducing the risk of disease transmission [8,12]. Additionally, the use of the BTT° method can enhance the precision of fever monitoring in pediatric patients, aiding in the prompt diagnosis and management of infectious diseases, including those with atypical presentations.

Body temperature (T_b_) measurement in children is important for the detection of pathological states such as fever and hypothermia [1]. The pulmonary artery temperature measurement method reflects the body’s core temperature; however, it is an invasive method, making it difficult to use in clinical practice [19]. There is no consensus on a non-invasive method for T_b_ measurement that can accurately predict the core temperature of the body in children. Batra et al. [19] have shown that the tympanic temperature (T_ty_) measurement method for children aged >2 years and the temporal artery temperature measurement method in all age groups are better than other non-invasive methods. Other studies [20,21] have shown the efficiency of the temporal artery temperature measurement method when compared to the tympanic and axillary thermometry method in predicting T_b_ in the pediatric emergency room. 

The axillary thermometry method is one of the oldest methods for recording temperature, and the tympanic method is currently the most used at home and in medical clinics [19]. These methods are more practical than other non-invasive methods [19]. Studies [21,22,23] have demonstrated that the tympanic method is as accurate as the axillary thermometry method; however, the studies suggest that clinicians should interpret peripheral thermometer readings with caution. Tympanic thermometry readings were also reported to be within the limits of agreement with axillary readings [24]. Systematics reviews that compared the axillary and tympanic measures described a lower correlation between these methods, as a small difference with wide confidence intervals was observed between measurements [25,26]. 

Dodd et al. [27] observed poor sensitivity when comparing the tympanic and axillary methods to measure temperature in 4098 children. Based on the combined estimates from multiple studies, the tympanic infrared thermometry method is ineffective at detecting fever in approximately 30% to 40% of febrile children. In these studies, fever was defined as a rectal temperature of 38 °C or higher. These findings suggest that the use of infrared ear thermometers may result in false negatives, failing to accurately identify the presence of fever in a significant percentage of children with fever. This is concerning, particularly in situations where precise fever detection is crucial, such as in medical or healthcare settings. This information reinforces the concern regarding the use of these thermometers in situations where the failure to detect fever can have serious implications, such as delaying appropriate treatment or allowing the spread of contagious diseases. This was a different result than that observed in the current study, given that a concordance between axillary and tympanic methods without proportional bias was described.

Limited data exists in the pediatric population regarding the use of the infrared thermography method to measure T_b_. Selent et al.’s study reported that the use of the infrared thermography (FLIR) method showed good accuracy in detecting fever in 855 children aged 6 months to 17 years when compared with traditional thermometry methods (e.g., rectal, oral, or axillary) in a pediatric emergency department. There is evidence that the temperature distribution in the skin of children aged <14 years follows a pattern similar to that described in adults. These authors revealed that the variation in T_b_ among children was lower than among adults, suggesting that infrared thermography could be a more precise diagnostic tool when used in pediatric patients than in the adult population [28]. According to these authors, the absence of any association with sex found in previous adult studies, might be explained by hormonal influences in adults that are not present in children.

Recent studies [29,30] have demonstrated that infrared thermography can be employed as a non-invasive technique for monitoring the temperature of pediatric patients in order to detect shock in its early stages. Although some studies have evaluated body temperature using an infrared thermography technique in children [29,30], no study has compared traditional T_b_ measurement methods with an infrared thermography technique that measures the temperature of the inner canthus of the eyes (T_ic,eye_) in the superomedial orbit (BTT°) in the pediatric population. The present data revealed a satisfactory agreement between the BTT° infrared thermography images and the reference methods (axillary and tympanic) for afebrile children. These findings suggest that BTT° temperatures obtained by infrared thermography is an interesting alternative for the measurement of T_b_ in children and can be used as a non-invasive method.

This study aimed to investigate the potential of BTT° temperature as a core temperature measure and assess its consistency and thermophysical features to identify a more precise method for the measurement of body temperature, particularly in pediatric cases. The BTT is a unique thermally transmissive and emissive pathway between the brain and the inner canthus of the eye, enabling the continuous and accurate measurement of brain temperature [7]. Its thermophysical properties are in harmony with the morphology, physiology, and physics of the brain, and the BTT works as a natural brain temperature indicator, serving no other purpose [7,31]. During the 2003 SARS outbreak, the BTT° was identified by a Yale University research group and was described as a thermophysical path providing a metric consistent with the Stefan–Boltzmann law of black body radiance [32]. Some authors have shown that the BTT° provides a consistent volumetric heat capacity, resulting in a more accurate assessment of fever from the hypothalamic thermoregulatory center [12]. This is particularly important during the COVID-19 pandemic, where suboptimal thermometry may result in the misdiagnosis of children. Fever is one of the main objective warning signs for COVID-19 and Kawasaki disease, and parental reliance on it can lead to false negatives or false positives, both of which can have serious consequences. By providing a more accurate measurement of body temperature, the infrared BTT° method has the potential to improve the detection of SARS-CoV-2 infections in children and reduce the risk of misdiagnosis [12]. However, despite the potential of infrared BTT° measurement to detect fever, a recent study has shown that average forehead (T_fh,avg_) and eye temperatures above the 37.5 °C fever threshold (T_fth_) are not sufficient to detect SARS-CoV-2 infection. Moreover, the study found that suspicious and confirmed SARS-CoV-2 infection cases with temperatures below the fever threshold (T_fth_) were only identified by a proposed algorithm that utilizes artificial intelligence and facial infrared imaging [8].

Despite the promising results of this study, and the rigorous methodology employed to obtain temperature measurements using infrared thermography in a controlled environment, some limitations should be noted. Firstly, the sample size was small, which leads to a low statistical power; thus, the data should be interpreted with caution due to the potential for inflated effect sizes. Secondly, a reference standard method to measure T_b_, such as rectal temperature, was not used to compare with the axillary and tympanic thermometry methods. Future research should also incorporate it to compare with the BTT° infrared thermography images.

Indeed, the validation of this technique as a simple, non-invasive method of T_b_ measurement in afebrile children could represent an advance for pediatric studies. The brain thermal tunnel (BTT°) temperature obtained by the infrared thermography method appears as an interesting alternative for T_b_ measurement in children. However, it should be noted that febrile children were not evaluated, thus precluding the inference of the reliability of using BTT° infrared thermography images to evaluate febrile children in the study.

## 5. Conclusions

The validation of the BTT° infrared thermography method as a reliable and accurate method for measuring body temperature in afebrile children has significant implications for public health. This non-invasive and efficient technique can improve the detection and monitoring of fever, aiding in the early identification and management of infectious diseases. The use of the BTT° method can play a crucial role in the context of disease outbreaks, such as COVID-19, by facilitating early case identification and reducing the risk of transmission. Furthermore, the application of the BTT° method in pediatric healthcare settings can lead to more precise temperature assessment, contributing to better clinical decision-making and improved patient outcomes. Overall, this study highlights the potential impact of the BTT° infrared thermography method on public health by offering a reliable and convenient tool for body temperature measurement in afebrile children and supporting early disease detection and intervention strategies.

However, it is important to recognize some of our study´s limitations. The research was carried out in a controlled indoor setting with specific conditions, including no solar radiation, a narrow range of air temperature, and low air velocity. Additionally, the test subjects were seated and unmoving when the measurements were being taken. As a result, even though our findings support the reliability of the BTT° method in this controlled environment, it is essential to understand that real-world situations, such as airports and outdoor settings, may introduce several variables (e.g., crowded and fast-moving groups of passengers, higher air velocity, increased metabolic heat production during brisk walking, and changes in clothing due to different seasons) that could affect skin and BTT temperatures. Therefore, it is imperative that these parameters be factored into a dedicated biothermodynamic adjustment formula, tailored to account for these specific conditions when contemplating the broader implementation of the BTT° method in such environments.

The results of this study provide evidence that the infrared thermography of brain tunnel temperature (BTT°) can be considered a reliable and accurate method for measuring body temperature in afebrile children. The high level of accuracy, which was comparable to the axillary and tympanic thermometry methods, 0.995 and 0.998, respectively, suggests that these methods can be used interchangeably with the BTT° method, depending on the availability of the equipment and the preference of the healthcare professional. However, it is important to note that further research is needed to validate and extend the current results. Future studies could explore the potential applications of this method in detecting and monitoring fevers in children, and in assessing the effectiveness of fever-reducing interventions. The use of the BTT° method could also be evaluated in different populations, such as febrile children or adults, to determine its potential advantages and limitations. Overall, this study contributes to the body of knowledge on non-invasive methods for measuring body temperature and highlights the potential of BTT° infrared thermography in clinical practice.

## Figures and Tables

**Figure 1 ijerph-20-06867-f001:**
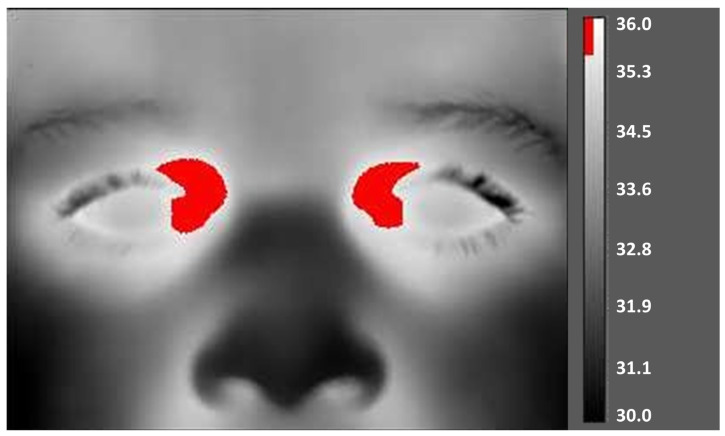
Representative image of the regions of interest (ROI) on the inner canthus of the eyes in the infrared thermal images (authors’ image). The temperature of the inner canthus of the eyes, T_ic,eye_ (located in the superomedial orbit) is represented in red, while the brain thermal tunnel is also shown.

**Figure 2 ijerph-20-06867-f002:**
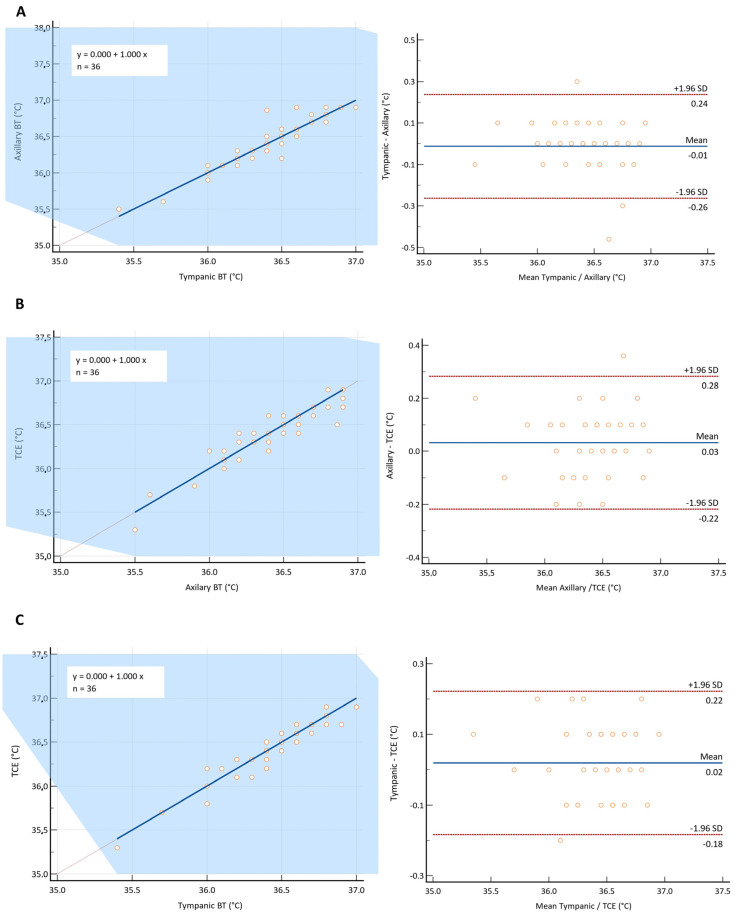
Passing–Bablok regression analysis and Bland–Altman plot analysis comparing different methods for body temperature (T_b_) measurement: (**A**) tympanic (T_ty_) versus axillary (T_ax_) thermometry; (**B**) axillary versus BTT° (temperature of the inner canthus of the eye (T_ic,eye_), brain thermal tunnel); (**C**) tympanic (T_ty_) versus BTT°. The dashed line in the Passing–Bablok regression analysis represents the line of identity, while the solid line represents the best fit. The dashed lines in the Bland–Altman plot indicate the upper and lower limit of agreement, and the solid line represents the average difference (bias).

**Table 1 ijerph-20-06867-t001:** General characteristics of the study population according to gender distribution.

	Females (n = 18)	Male (n = 18)	*p* Value
Overweight/Obesity (%)	22	28	0.700
BMI (kg/m^2^)	16.7 [3.3] (14.5–28.2)	16.6 [3.1] (14.7–26.5)	0.988
SBP (mmHg)	95 [7.0] (89–105)	96 [8.0] (90–110)	0.293
DBP (mmHg)	58 [7.0] (51–69)	59 [8.0] (54–72)	0.988
Axillary, T_ax,b_ (°C)	36.4 [0.4] (35.5–36.9)	36.5 [0.4] (35.9–37.0)	0.650
Tympanic, T_ty,b_ (°C)	36.4 [0.4] (35.5–36.8)	36.4 [0.5] (35.9–37.0)	0.628
BTT°, T_ic,eye_ (°C)	36.4 [0.3] (35.4–36.8)	36.5 [0.6] (35.6–36.9)	0.372

Values expressed as percentage, median, [interquartile range] or (minimum–maximum values). T_b_: Body Temperature; BMI—Body Mass Index; SBP—Systolic Blood Pressure; DBP—Diastolic Blood Pressure; BTT°: Temperature of the inner canthus of the eye, T_ic,eye_ (brain thermal tunnel).

**Table 2 ijerph-20-06867-t002:** Concordance correlation coefficients between the axillary, tympanic, and BTT° methods, precision (Pearson ρ), and accuracy (bias correction factor, Cb).

		CCC (95% CI)	Precision ρ	Accuracy C_b_
Axillary (°C)	Tympanic (°C)	0.93 (0.87 to 0.96)	0.930	0.997
	BTT° (°C)	0.93 (0.86 to 0.96)	0.930	0.995
Tympanic (°C)	BTT° (°C)	0.94 (0.90 to 0.97)	0.951	0.998

Abbreviations: BTT°: Brain Thermal Tunnel Temperature, CCC: Concordance Correlation Coefficient, 95% CI: 95% Confidence Intervals.

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
