# Peer review of "Infrared Imaging of the Brain-Eyelid Thermal Tunnel: A Promising Method for Measuring Body Temperature in Afebrile Children"

_ijerph, 2023, doi:10.3390/ijerph20196867_

Round 1
Reviewer 1 Report
This article describes the issues related to Infrared Imaging of the Brain Eyelid Thermal Tunnel: A Pro mising Method for Measuring Body Temperature in Afebrile Children. In their considerations, the authors presented the previously used methods of the temperature detection among children, to whom they applied the proposed method. In the work, the authors correctly presented the issues related to the previous research and proposed their approach to the analyzed topic. The presented issues are up-to-date and undergo the constant development, especially with the increasing efforts to effectively and quickly detect changes in the body temperature that indicate the pathological changes. Based on the correctly presented analysis of the results of the research, the complete conclusions were drawn. The reviewed work is well organized with an introduction, the theoretical part, the presentation of simulation results and the conclusions. After the theoretical part, the study presents the results of the research and their analysis. Each part is correctly presented and correlated with the rest of the article.
Recommendations for improving the manuscript:
1. The purpose of the work presented at the beginning of the article should be clarified (which is the work of the authors).
2. It is not explained how the thermometers were calibrated.
3. As the authors noted, the research work was carried out on a small number of populations. Is it sufficient to draw the presented conclusions?
4. Is it possible to meet the requirements related to thermographic measurements (constant ambient parameters, acclimatization time) in real conditions?
5. Did the authors try to conduct at least a single study on people with diagnosed increased temperature?
6. What benefits and to whom can the results presented in the article bring in the future?
Author Response
Thank you for taking the time to review our manuscript. We greatly appreciate your feedback and suggestions, which have been carefully considered and incorporated into the revised version of the article. Your contributions have undoubtedly helped to improve the quality and accuracy of our work. Thank you again for your diligent review and constructive input.
- The purpose of the work presented at the beginning of the article should be clarified (which is the work of the authors).
We appreciate your feedback. We have taken every measure to ensure that the purpose of our research is transparent and understandable to readers.
As the authors of this article, our primary objective is to investigate the utility of brain thermal tunnel (BTT°) infrared thermography as a means of assessing body temperature in afebrile children. This objective has been explicitly articulated at various points within the manuscript to ensure clarity.
In the abstract, we state: “This preliminary study aimed to evaluate the concordance between the temperature of the inner canthus of the eye (Tic,eye), referred to as the brain-eyelid thermal tunnel (BTT°) with axillary and tympanic methods in afebrile children.”
In the introduction, we further emphasize our goal: “This study aimed to assess the correlation between brain thermal tunnel (BTT°) infrared thermography and axillary (Tax) and tympanic (Tty) temperatures in afebrile children.”
Throughout the discussion section, we consistently reiterate our study's intent: “Our study aimed to investigate the potential of the BTT° as a measure of core temperature and to evaluate its consistency and thermophysical features to provide a more precise measurement of body temperature, particularly in children.”
We appreciate the reviewer's attention to this matter and are confident that these clarifications adequately address their concern.
- It is not explained how the thermometers were calibrated.
We appreciate your valuable feedback and are grateful for the opportunity to address this important aspect of our study. We have reviewed and added these details to the text. In our study, we employed a meticulous calibration process for all thermometers used, including the Brain Thermal Tunnel (BTT°) infrared thermography device, axillary thermometer (Tax), and tympanic thermometer (Tty). We added your suggestion and explained in red in the text how we calibrate the axillary and tympanic thermometers.
The calibration of the Brain Thermal Tunnel (BTT°) infrared thermography device was explained in the Methods section: The thermal imager was tested using high-quality calibration sources called black bodies. These black bodies had calibrated radiance temperatures with an uncertainty of no more than 0.1 °C (with a confidence level of approximately 95%) and stability better than ±0.002 °C. The calibration process was carried out by a competent laboratory specialized in radiation thermometry and following international measurement standards. The service centers involved in the calibration were certified according to ISO 9001, and the temperature reference standards were traceable to either the SP Technical Research Institute of Sweden or the National Institute of Standards and Technology (NIST) in the United States. To calibrate the thermal imager, radiation sources were used that could be traced back to the National Standards at RISE, Research Institutes of Sweden, and NIST. The black body sources used had a range of radiance temperatures and control intervals suitable for laboratory testing as per the specified standard. These black bodies had a known emissivity greater than 0.995. The aperture size of the black body source was sufficiently large to ensure that it did not affect the thermal imager's temperature measurement and allow for clear identification of color changes at the workable plane.
- As the authors noted, the research work was carried out on a small number of populations. Is it sufficient to draw the presented conclusions?
Thank you for your insightful comment regarding the sample size of our study. We appreciate your diligence in assessing our research. As discussed in the "Discussion" section of our manuscript, we acknowledge that the sample size in our study was relatively small. This limitation indeed affects in part the statistical power of our findings. However, it was highlighted in the text that a smaller sample size can lead to inflated effect sizes and should be considered when interpreting our conclusions.
While our study provides valuable insights into the potential of the thermographic method, we recognize that a larger and more diverse sample would further strengthen the robustness of our conclusions. We view our research as a foundational step for children's temperature screening, and we encourage future studies with larger cohorts to corroborate our findings and enhance the generalizability of our results.
We agree with your assessment, and we have addressed this limitation in our discussion. We remain committed to advancing this research area and appreciate your feedback, which will undoubtedly contribute to its refinement.
- Is it possible to meet the requirements related to thermographic measurements (constant ambient parameters, acclimatization time) in real conditions?
We appreciate your interest in the practical applicability of our thermographic measurement methods in real-world scenarios. We understand your concern, and we'd like to provide further clarification.
While our study was conducted in a controlled environment with carefully maintained ambient conditions, it's important to emphasize that the controlled conditions were designed to mimic the most common real-world scenarios as closely as possible. In essence, our approach aimed to bridge the gap between ideal laboratory settings and the complexities of everyday situations.
In the real world, achieving constant ambient parameters and extended acclimatization times can be challenging due to various environmental factors. However, we believe our study's controlled conditions represent a best-case scenario, demonstrating the potential of our thermographic method under favorable circumstances. This is a crucial first step in establishing the method's viability for broader real-world applications.
In practice, it may not always be feasible to replicate these controlled conditions precisely. Still, our results provide valuable insights into the method's performance and its adaptability to different environments. We acknowledge that variations in ambient conditions may occur, but our study forms a foundational basis upon which further research and real-world validation can build.
We hope this clarifies that while our study was conducted under controlled conditions; it serves as a proof of concept and demonstrates the method's potential for real-world applicability. We remain committed to advancing our research in this area and exploring the method's robustness across diverse settings.
- Did the authors try to conduct at least a single study on people with diagnosed increased temperature?
Thank you for your question regarding whether we conducted a study involving people with diagnosed increased temperature. We indeed conducted a recent study involving febrile and subfebrile individuals and published our findings in a paper titled "Infrared image method for possible COVID-19 detection through febrile and subfebrile people screening" (Brioschi ML, Dalmaso Neto C, Toledo M, Neves EB, Vargas JVC, Teixeira MJ, J Therm Biol. 2023 Feb;112:103444 – doi: 10.1016/j.jtherbio.2022.103444. Epub 2022 Dec 28. PMID: 36796899; PMCID: PMC9794388). This study explored the use of infrared thermography in the screening of individuals with increased body temperatures based on a similar methodology to our actual paper. We appreciate your interest in our research, and we hope this information addresses your query.
- What benefits and to whom can the results presented in the article bring in the future?
Thank you for your query about the potential benefits of our study's results in the future to improve the clarity and readability of our paper. Our study focused on validating the use of BTT° (brain thermal tunnel) infrared thermography as a reliable and non-invasive method for measuring body temperature in afebrile children. We have highlighted the significance of our findings in the context of public health, particularly during infectious disease outbreaks like COVID-19.
Our study demonstrates that BTT° provides temperature measurements that are concordant with traditional axillary and tympanic methods without proportional bias. This establishes BTT° as a valuable alternative for measuring body temperature in children, with implications for various settings, including schools, public transport, and healthcare facilities. Rapid and accurate temperature assessment using BTT° can aid in the early identification of potential cases of fever, enabling timely isolation, testing, and treatment, thereby reducing the risk of disease transmission.
Furthermore, the precision of BTT° in measuring fever in pediatric patients can enhance the diagnosis and management of infectious diseases, particularly those with atypical presentations. In contrast to other non-invasive methods, BTT° has shown promise in providing a more accurate assessment of fever, which is crucial during the COVID-19 pandemic, where suboptimal thermometry can lead to misdiagnosis.
While we acknowledge the limitations of our study, including the sample size and the need for further research, we believe that our findings offer a significant contribution to the field of non-invasive temperature measurement. Our study highlights the potential impact of BTT° infrared thermography on public health by providing a reliable and convenient tool for body temperature measurement in afebrile children and supporting early disease detection and intervention strategies.
Once again, we appreciate your valuable feedback, which has helped us improve the quality and clarity of our manuscript. We hope that these revisions address your concerns and enhance the overall quality of our paper.

Reviewer 2 Report
The study solidifies the brain-eyelid thermal tunnel method (BTT°) as a dependable approach for measuring body temperature in afebrile children. BTT° demonstrates strong concordance with conventional methods like axillary and tympanic thermometry. These findings carry significant implications for public health, particularly during outbreaks such as COVID-19. BTT° offers a non-invasive and accurate alternative for measuring children's body temperature.
While the study holds significance, there are some aspects that may require attention and refinement.
1. In the abstract “The results revealed excellent concordance between the tympanic, axillary, and BTT° methods”. Please consider replacing "between" with "among."
2. In the discussion section, there are statements lacking proper references, such as "Based on the combined estimates from multiple studies." It is recommended to provide specific references for such statements to strengthen the credibility of the discussion.
3. In the results description section, it would be beneficial to clarify the rationale behind conducting certain experiments. Consider adding explanatory sentences like "To determine..." and "We conducted..." to elaborate on the purpose and methodology of specific experiments. This will provide readers with a better understanding of the study's approach and its outcomes.
4. Figure 2 requires reorganization with larger font sizes and higher-resolution images for enhanced readability and visual impact.
5. The description of the method raises questions about the practicality of using infrared thermography for mass screening. While the abstract and discussion highlight its applicability for mass screening in various settings, does the method section describe measuring children one by one in a seated position? The authors should carefully revise this description to ensure consistency and clarify how the method can be applied for mass screening. The advantages of this approach should be articulated more explicitly. This will enhance the overall coherence of the paper and strengthen the conclusions drawn from the research.
Minor editing of English language required.
Author Response
The study solidifies the brain-eyelid thermal tunnel method (BTT°) as a dependable approach for measuring body temperature in afebrile children. BTT° demonstrates strong concordance with conventional methods like axillary and tympanic thermometry. These findings carry significant implications for public health, particularly during outbreaks such as COVID-19. BTT° offers a non-invasive and accurate alternative for measuring children's body temperature.
While the study holds significance, there are some aspects that may require attention and refinement.
Thank you for taking the time to review our manuscript. We greatly appreciate your feedback and suggestions, which have been carefully considered and incorporated into the revised version of the article. Your contributions have undoubtedly helped to improve the quality and accuracy of our work. Thank you again for your diligent review and constructive input.
- In the abstract “The results revealed excellent concordance between the tympanic, axillary, and BTT° methods”. Please consider replacing "between" with "among."
We have made the corrections, thank you. Your feedback and suggestions have been incorporated into the revised abstract.
- In the discussion section, there are statements lacking proper references, such as "Based on the combined estimates from multiple studies." It is recommended to provide specific references for such statements to strengthen the credibility of the discussion.
Thank you for your valuable feedback on our manuscript. We have addressed your concern by including a reference (Dodd et al. [23]) to support the statement regarding the inefficacy of tympanic infrared thermometry in detecting fever in a significant percentage of febrile children. This addition enhances the credibility of our discussion section by providing a specific source for the information presented.
Furthermore, we have provided context for this information, explaining how it differs from the results of our study. This clarification helps to illuminate any potential discrepancies or divergences in the findings.
As a result, we believe that our correction aligns with your request by furnishing a specific reference to bolster the statement in the discussion section, ultimately strengthening the overall credibility of our article.
- In the results description section, it would be beneficial to clarify the rationale behind conducting certain experiments. Consider adding explanatory sentences like "To determine..." and "We conducted..." to elaborate on the purpose and methodology of specific experiments. This will provide readers with a better understanding of the study's approach and its outcomes.
Thank you for your constructive feedback on our manuscript. Your insightful suggestion to clarify the rationale behind specific experiments in the 'Results' section has been extremely helpful. We have carefully considered your feedback and have included in red explanatory sentences that aim to provide a more comprehensive understanding of the purpose and methodology of the experiments in question. With these additions, we hope to enhance the clarity and transparency of our study's approach and results.
Once again, we sincerely appreciate your diligent review and thoughtful recommendations.
- Figure 2 requires reorganization with larger font sizes and higher-resolution images for enhanced readability and visual impact.
Thank you for your suggestion to improve the readability and visual impact of Figure 2. We have revised the figure as per your recommendation, increasing the font sizes and providing higher-resolution images. These changes should enhance the overall clarity and effectiveness of Figure 2 in conveying our research findings.
- The description of the method raises questions about the practicality of using infrared thermography for mass screening. While the abstract and discussion highlight its applicability for mass screening in various settings, does the method section describe measuring children one by one in a seated position? The authors should carefully revise this description to ensure consistency and clarify how the method can be applied for mass screening. The advantages of this approach should be articulated more explicitly. This will enhance the overall coherence of the paper and strengthen the conclusions drawn from the research.
We appreciate your feedback and have carefully revised the Methods section of our manuscript to provide a more detailed explanation of the practicality and applicability of using infrared thermography for mass screening.
We have included comprehensive information about the experimental setup, controlled conditions, participant positioning, and equipment calibration to address your concerns. Additionally, we highlighted the meticulous calibration of the tripod system, the synchronization of the thermographic camera's image acquisition with the child's natural blinking pattern, and the careful alignment of the camera's lens. These details emphasize the reliability and practicality of our method for mass screening.
We believe that these additions enhance the overall coherence of the paper and provide a clearer understanding of how our method can be effectively applied in various settings. Thank you for your valuable input, which has helped us improve the quality and clarity of our manuscript.
Thank you again. Your expertise and guidance are greatly appreciated.

Reviewer 3 Report
I thank the authors for the answers to the reviewers comments and impriving the manuscript.
But there are still a few concerns with the new version of the manuscript, such as:
- Keywords should be in alphabetical order
- I strongly suggest authors from refraining using personal pronouns such as "we" and "our" throughout the text and I encourage them to write it in an impersonal form of writing.
- At the thermal camera description there is no such thing as pixels, it should be the Focal Plain Array sensor size of 320x240, pixels are related to the final image where a sensor can correspond to a pixel (without interpolation) or more than one pixel (with interpolation)
- Also, thermal sensitivity is not the same thing as Non-Equivalent Temperature Difference (NETD), in physics those concepts are not the same thing, some camera manufacturers mislead this in the equipment description, in this case the correct value corresponds to a NETD
- I was not convinced with the answer provided by the authors concerning the reliability of the measured ROI
- The brain thermal tunnel (BTT) concept is controversial concept and not widely accepted among scientific community and had some bizarre links in the past with some onclogical conditions, it is proposed by a Brazilian scholar at Yale and strangely has 3 references in this manuscript, I have the suspition of an attempt of overcitation to this author by this manuscript authors.
I strongly suggest authors from refraining using personal pronouns such as "we" and "our" throughout the text and I encourage them to write it in an impersonal form of writing.
Author Response
I thank the authors for the answers to the reviewers comments and impriving the manuscript.
Thank you for your valuable feedback on our manuscript. We appreciate your time and effort in reviewing our work. Your insights have been carefully considered, and we are pleased to confirm that your suggestion has been incorporated into the revised version of the manuscript. Your contributions have undoubtedly enhanced the quality and accuracy of our article. Once again, we appreciate your diligent review and constructive input.
But there are still a few concerns with the new version of the manuscript, such as:
- Keywords should be in alphabetical order
We appreciate the observation and have corrected it in the text.
- I strongly suggest authors from refraining using personal pronouns such as "we" and "our" throughout the text and I encourage them to write it in an impersonal form of writing.
We appreciate the observation and have corrected it in the text.
- At the thermal camera description there is no such thing as pixels, it should be the Focal Plain Array sensor size of 320x240, pixels are related to the final image where a sensor can correspond to a pixel (without interpolation) or more than one pixel (with interpolation)
We have carefully considered all your comments, particularly the concern regarding the Focal Plain Array sensor size and interpolation. Your observations were keen, and we fully acknowledge the importance of this aspect in our research. As per your guidance, we have made the necessary revisions to the text, which is now highlighted in red throughout the manuscript.
- Also, thermal sensitivity is not the same thing as Non-Equivalent Temperature Difference (NETD), in physics those concepts are not the same thing, some camera manufacturers mislead this in the equipment description, in this case the correct value corresponds to a NETD.
Sorry for the misunderstanding. We agree and we changed it in the text. Non-equivalent Temperature Difference (NETD) is a crucial performance metric in thermal imaging equipment, representing its ability to detect small temperature variations. In contrast to thermal sensitivity, NETD takes into account the non-uniformity and variations in the sensor's response across the image, providing a more accurate measure of the camera's performance.
- I was not convinced with the answer provided by the authors concerning the reliability of the measured ROI - I am not happy with the answer to the ROI question and the provided answer makes me believe that the authors were not cautious in recording the data.
We appreciate the reviewer's diligence in assessing the reliability of our measured ROI. In our study, great care was taken to ensure the precision and consistency of the ROI selection process.
After capturing the infrared thermography images, we took meticulous steps to ensure the precision and consistency of ROI (Region of Interest) selection, particularly focusing on the inner canthus of the eyes, denoted as Tic,eye. To accomplish this, we employed the ellipse tool available within the FLIR ResearchIR Max software. It's important to note that this software tool is not only widely recognized for its reliability but is also recommended by the manufacturer of the same sensor used in our study for ROI analysis.
We want to emphasize that this method is a standard practice followed by numerous other researchers in the field of infrared thermography. Its acceptance and adoption by the broader scientific community highlight its reliability and validity in thermographic analysis. This methodology has been consistently proven effective in various research studies, reinforcing our confidence in its suitability for our investigation.
Furthermore, it is of utmost importance to emphasize that the determination of the maximal BTT° values on both sides (Tic,eye,max) was exclusively performed by a certified researcher who possesses extensive experience and expertise in the field of infrared thermography. This researcher, identified in the text as F.D.M (Franciele De Meneck), is not only one of the authors but has also published numerous studies employing a similar methodology.
Her qualifications and extensive training in infrared thermography ensure the precise and reliable execution of these measurements. F.D.M.'s track record of previous research utilizing this methodology further underlines her competence in conducting such measurements accurately, reaffirming the credibility of our data.
For clarity, we have included a representative image in Figure 1, which illustrates the regions of interest (ROI) on the inner canthus of the eyes in our infrared thermal images. In this image, the temperature of the inner canthus of the eyes (Tic,eye), situated in the superomedial orbit, is depicted in a red region (isotherm), along with the visualization of the BTT representative region.
This study is the outcome of F.D.M.'s doctoral thesis. Additionally, we took meticulous precautions in various aspects of the study, including the positioning of the thermographic camera. To ensure precise temperature measurements of the inner canthus of the radiant eyelid on children's faces, we utilized a meticulously calibrated tripod system equipped with built-in leveling mechanisms. The thermographic camera operator positioned the camera perpendicular to the child's face and made meticulous adjustments to align the camera's lens accurately. The operator received training to maintain this alignment throughout the data collection process by monitoring the camera's position and making real-time adjustments when necessary. This rigorous approach was essential to minimize measurement errors resulting from angles between the camera and the target area, thereby ensuring temperature measurement reliability. Additionally, a specialized trigger mechanism synchronized the thermographic camera's image capture with the child's natural blinking pattern. This synchronization ensured that measurements were taken during moments when the child's eyes were stable and relaxed, reducing the potential for artifacts or inaccuracies in the data. The trigger mechanism underwent careful calibration to avoid any disruption to the child's comfort or natural behavior during the measurements.
We hope this additional information reassures the reviewer regarding the meticulousness and reliability of our ROI selection and measurement process.
- The brain thermal tunnel (BTT) concept is controversial concept and not widely accepted among scientific community and had some bizarre links in the past with some onclogical conditions, it is proposed by a Brazilian scholar at Yale and strangely has 3 references in this manuscript, I have the suspition of an attempt of overcitation to this author by this manuscript authors.
We appreciate your thorough review of the manuscript discussing the Brain Thermal Tunnel (BTT) concept. Your insights are valuable, and we would like to address your concerns regarding the controversy surrounding this concept and its association with Yale University.
It is essential to acknowledge that science, by its very nature, thrives on controversy and debate. The pursuit of knowledge often involves the exploration of unconventional ideas and hypotheses. The fact that the BTT concept has not gained widespread acceptance within the scientific community is not surprising and it's also not unworthy at all. The controversy can lead to new discoveries. It is precisely this environment of skepticism and inquiry that drives researchers to investigate topics like BTT, to raise new elements that either substantiate or refute these hypotheses.
Regarding the inclusion of references from Yale University´s authors, it is important to clarify that such references do not necessarily imply overcitation. Rather, they may serve several purposes, such as providing context, supporting specific claims, or highlighting relevant prior work about a new topic with a few references. Therefore, references were selected based on their relevance to the manuscript's content. The inclusion of these should be evaluated in the context of the manuscript's content and its contribution to the field.
Furthermore, it is common in scientific research to draw upon the expertise of professionals and researchers who are considered authorities in their respective fields, regardless of their scholar status. The fact that these professionals are engaged in the investigation of the BTT concept lends credibility to its exploration.
Lastly, the affiliation with Yale University is indeed noteworthy, as it is known for promoting innovative research across different fields. It is no surprise that researchers affiliated with such prestigious institutions often come up with groundbreaking ideas. The research related to BTT is serious, regardless of the controversies that may exist.
We understand and appreciate the concerns regarding the controversy surrounding the BTT concept and the references mentioned in the manuscript. However, we firmly believe that further research and debate are crucial for advancing our understanding of this concept and its possible implications. We encourage everyone to carefully consider the manuscript's content and its contributions to the scientific community in a broader context.
Comments on the Quality of English Language
I strongly suggest authors from refraining using personal pronouns such as "we" and "our" throughout the text and I encourage them to write it in an impersonal form of writing.
We appreciate the observation and have corrected it in the text.
Once again, thank you for your invaluable contributions to this project.
Your expertise and guidance are greatly appreciated.

Reviewer 4 Report
Dear Authors,
Thank you for addressing some of my previous comments.
However, in the latest manuscript version I detected that the following concerns were not addressed yet:
- The thermographer (operator, researcher handling the thermographic camera) cannot stay perpendicular to the object surface (children faces). Please expand on this in your methodology.
- The Authors did not specify the thermography triggering system.
Actually, I expected red writing on Section 2. Methods.
Author Response
Dear Authors,
Thank you for addressing some of my previous comments.
However, in the latest manuscript version I detected that the following concerns were not addressed yet:
- The thermographer (operator, researcher handling the thermographic camera) cannot stay perpendicular to the object surface (children faces). Please expand on this in your methodology.
- The Authors did not specify the thermography triggering system.
Actually, I expected red writing on Section 2. Methods.
We have carefully considered all your comments, particularly the concern regarding the positioning of the thermographic camera operator in relation to the child's face and the thermography triggering system. Your observations were keen, and we fully acknowledge the importance of this aspect in our research. As per your guidance, we have made the necessary revisions to the text, which is now highlighted in red throughout the manuscript.
We wanted to extend our heartfelt gratitude for your time and effort in reviewing our manuscript. Your feedback has been invaluable in helping us to improve the quality and clarity of our work. We´re truly grateful for your insights and expertise, and we appreciate the care and attention you have given to our project.
Thank you again for your support and guidance.
Your expertise and guidance are greatly appreciated.

Round 2
Reviewer 2 Report
Thank you to the authors for addressing my concerns and for the revised manuscript. I no longer have any significant comments to make. Consequently, I am pleased to recommend the acceptance of this paper on 'Infrared Imaging of the Brain-Eyelid Thermal Tunnel' for publication.
Minor editing of English language required.
Author Response
We sincerely appreciate the time and effort you invested in reading through our manuscript, and we are happy to learn that the revisions successfully addressed your concerns. Your suggestions have greatly improved the caliber of our work.
Reviewer 3 Report
Thank you for addressing the reviewers comments and suggestions, it improved significantly the manuscript.
Although, since distance and angle have a strong influence in the measurement, a bibliographic reference is missing, please add: Vardasca, R. et al. (2017). The influence of angles and distance on assessing inner-canthi of the eye skin temperature. Thermology international, 27(4), 130-135.
Author Response
Thanks for your suggestion. We added the reference in the text.
"The thermographic camera operator was positioned perpendicular to the child's face plan (90⁰) and meticulously adjusted the tripod to align the camera's lens accordingly [14]"
Reviewer 4 Report
There are just two minor changes that have significant importance on the methodology, based on my previous review comments.
I understand Authors may have overlooked some nuances in the review comments concerning the position of the thermal camera operator and the implications of it.
"The thermographic camera operator was positioned perpendicular to the child's face" should be (of course, in lower case):
"The thermographic camera operator was positioned NON-perpendicular (UP TO 60º) to the child's face"
and:
"This meticulous approach was crucial to minimize measurement errors caused by oblique angles between the camera and the target area" should be:
"This meticulous approach was crucial to minimize measurement errors caused by EXCESSIVE oblique angles between the camera and the target area, AND THE THERMAL RADIATION FROM THE OPERATOR WHEN THEY ARE POSITIONED PERPENDICULAR TO TARGET SURFACES"
The following papers could be included in your manuscript for reference to support the explanation of the directional emissivity, which affects the surface temperature:
https://doi.org/10.1115/IMECE2012-88105
https://doi.org/10.3390/sym13020335
Nothing new
Author Response
Thanks for your suggestion. However, we believe that there are different ways of describing the positioning of the camera in relation to the patient, with angle measurements starting in different references, which can cause confusion. The reference we used, which was also recommended by Reviewer 3, describes the patient's positioning as presented in the text. I also attach a figure presented by the reference used that illustrates this position, to better clarify the case.
“The thermal camera operator was positioned perpendicular to the child's facial plane (90⁰) and meticulously adjusted the tripod to align the camera lens accordingly [14]”
